# DamID identifies targets of CEH-60/PBX that are associated with neuron development and muscle structure in *Caenorhabditis elegans*

**Pieter Van de Walle**[1], **Celia Muñoz-Jiménez**[2], **Peter Askjaer**[2], **Liliane Schoofs**[1], **Liesbet Temmerman**[1]*

**1** Animal Physiology and Neurobiology, University of Leuven (KU Leuven), Leuven, Belgium, **2** Andalusian Center for Developmental Biology (CABD), CSIC/JA/Universidad Pablo de Olavide, Seville, Spain

\* Liesbet.Temmerman@kuleuven.be

**Data Availability Statement:** Raw and processed DamID data are publicly available at ArrayExpress (accession E-MTAB-9539).

## Abstract

Transcription factors govern many of the time- and tissue-specific gene expression events in living organisms. CEH-60, a homolog of the TALE transcription factor PBX in vertebrates, was recently characterized as a new regulator of intestinal lipid mobilization in *Caenorhabditis elegans*. Because CEH-60's orthologs and paralogs exhibit several other functions, notably in neuron and muscle development, and because *ceh-60* expression is not limited to the *C. elegans* intestine, we sought to identify additional functions of CEH-60 through DNA adenine methyltransferase identification (DamID). DamID identifies protein-genome interaction sites through GATC-specific methylation. We here report 872 putative CEH-60 gene targets in young adult animals, and 587 in L2 larvae, many of which are associated with neuron development or muscle structure. In light of this, we investigate morphology and function of *ceh-60* expressing AWC neurons, and contraction of pharyngeal muscles. We find no clear functional consequences of loss of *ceh-60* in these assays, suggesting that in AWC neurons and pharyngeal muscle, CEH-60 function is likely more subtle or redundant with other factors.

## 1. Introduction

PBC-class transcription factors fulfill a wide range of developmental functions in many organisms (reviewed in [1]). In the nematode *Caenorhabditis elegans*, they are represented by CEH-20, -40 and -60. While CEH-20 and -40 have been extensively characterized as drivers of neuronal, muscle and general mesodermal development [2–8], CEH-60 remained poorly understood. Recently, it has been shown that CEH-60, like other PBC-class proteins, interacts with a conserved partner UNC-62 and that this interaction occurs in the adult intestine to control lipid mobilization through yolk protein production [9, 10]. It is also argued that the default mode of action of CEH-60 is as a repressor of transcription. This is supported by the fact that in absence of functional CEH-60, ~75% of differential transcripts are upregulated vs ~25% downregulated and that upregulated genes associate more closely with CEH-60 binding sites than downregulated genes [9].

**Funding:** PVdW is an n SB PhD fellow of the FWO Flanders (1S00617N, https://www.fwo.be). The authors are grateful to the FWO Flanders (G095915) and KU Leuven (https://www.kuleuven.be, C16/19/003), the Spanish Ministry of Science and Innovation (PID2019-105069GB-I00 and MDM-2016-0687 to PA) and the Spanish Research Council (2019AEP142 to PA) for financial support. We are especially grateful to the Genie COST action (BM1408) for supporting this research.

**Competing interests:** The authors have declared that no competing interests exist.

The role of CEH-60 may not be limited to its adult-specific function in lipid mobilization, as CEH-60 activity is also observed in a specific pair of sensory neurons, identified as AWC ("Amphid Wing C"), and in the pharyngeal muscle cells PM6 [10, 11]. AWC neurons are a pair of ciliated neurons in the head region of the animal, their neuron bodies residing near the nerve ring that forms the nexus of the *C. elegans* nervous system [12]. The best-characterized function of AWC neurons is sensing volatile odors that may signal the presence of food in natural environments to regulate movement towards these odors, termed chemotaxis [13]. To a lesser degree, AWC neurons are involved in sensing temperature [14] and electric fields [15]. Pharyngeal muscles are involved in food intake, and can thus regulate metabolism [16].

Other PBX proteins in *C. elegans*, CEH-20 and CEH-40, as well as their vertebrate counterparts, have well-characterized roles in neuronal development [17–19] and muscle development [20–24]. Because of *ceh-60*'s functionally orphan neuronal and pharyngeal expression pattern and involvement of its closest relatives in numerous other developmental processes, we decided to look for direct gene targets of CEH-60 using DNA adenine methyltransferase identification (DamID).

DamID is a technique to map the interactions between a protein of interest (POI) and the genome. It has been successfully performed in *Drosophila melanogaster* [25], mammalian cell lines [26] and *C. elegans* [27]. This technique was originally developed to characterize the interactions of chromatin proteins with DNA [25], but can also be used to find target genes of transcription factors [28–30]. In this paper, DamID is used to map the interactions between the transcription factor CEH-60 and the promoter regions of its gene targets. Compared to immunoprecipitation techniques such as ChIP-seq, DamID has the advantage of providing an accumulated signal of protein binding instead of a "snapshot".

DamID relies on the fusion of a POI to a Dam domain, which will methylate the adenine of GATC sites in the genome that are spatially close to where the fusion protein interacts with the DNA [25]. Because GATC methylation does not occur naturally in eukaryotes, the methylation sites can be used as markers for sites of POI-DNA interaction.

In this study, we use DamID to find the genomic targets of CEH-60 and classify them. Genes linked to neuronal development, along with muscle structure genes, indeed emerge as some of the strongest target candidates. We subsequently investigate the morphology and sensory function of AWC neurons as well as pharyngeal pumping in *ceh-60* mutants. In these assays we find no clear differences between mutants and controls, suggesting that the function of CEH-60 in neurons and muscle cells may be more subtle or redundant with other factors.

## 2. Materials and methods

### 2.1 Strains and culture methods

*C. elegans* were grown under standard conditions [31], fed *Escherichia coli* OP50, and raised at 20˚C. For details on strain names and genotypes, see Table 1.

### 2.2 DamID: Plasmids and strains

To obtain tissue-specific control over the *dam*::*ceh-60* transgene, we used a FLP-based gene expression toolkit [32]. In the experimental DamID strain LSC1724, an inducible heat-shock promoter is followed by a fusion of an *mCherry* gene with a fragment encoding the histone HIS-58, flanked by FRT sites, integrated on chromosome II. This FRT-excisable reporter cassette is followed by a fusion gene of *ceh-60* and *dam*. Additionally, integrated on chromosome IV, this strain contains a *ceh-60* promoter driving expression of the flippase FLP, followed by an *mNeonGreen* reporter gene. Another strain, LSC1722, in which the *dam*::*ceh-60* fusion gene

**Table 1. List of *C. elegans* strains used, including genotype and source.**

| Strain | genotype | source/reference |
|---|---|---|
| BN578 | *unc-119(+) lmn-1p::mCherry::his-58 II;* | [32] |
| | *unc-119(+) myo-2p::gfp IV* | |
| PY2417 | *oyIs44 [odr-1p::rfp + lin-15(+)]* | *Caenorhabditis* Genetics Center, University of Minnesota, MN, USA |
| PR891 | *osm-1(p816) X* | |
| CB1124 | *che-3(e1124) I* | |
| LSC897 | *ceh-60(lst466) X* | [11] |
| LSC903 | *ceh-60(lst491) X* | |
| BN812 | *unc-119(+) lmn-1p::mCherry::his-58 II;* | This study |
| | *unc-119(+) ceh-60p::FLP::SL2::mNG IV* | |
| LSC1689 | *unc-119(+); hsp16.41p::FRT::mCherry::* | |
| | *his-58::FRT::dam::ceh-60 II* | |
| LSC1722 | *unc-119(+); hsp16.41p::FRT::mCherry::his-58::FRT::gfp::dam II;* | |
| | *unc-119(+) ceh-60p::FLP::SL2::mNG IV* | |
| LSC1724 | *unc-119(+) hsp16.41p::FRT::mCherry::his-58::FRT::dam:: ceh-60 II; unc-119(+) ceh-60p::FLP::SL2::mNG IV* | |
| LSC1863 | *oyIs44 [odr-1p::rfp + lin-15(+); ceh-60(lst466)* | |
| LSC1878 | *eat-2(ad465) II;* | Gift of Brecht Driesschaert and Lucas Mergan, KU Leuven, Belgium |
| | *lstIs24[unc-122p::DsRed::unc-54 3' UTR]* | |

mNG = mNeonGreen.

is replaced with *gfp*::*dam*, serves as the control for aspecific DamID signal (= noise) during analysis.

Only in tissues in which FLP is expressed, the FRT-flanked cassette on chromosome II is excised, inactivating the *mCherry* and *his-58* reporters, and ceding heat-shock control to the *dam*::*ceh-60* or *gfp*::*dam* fusion genes. In this way, Dam fusion proteins are only expressed in tissues in which the flippase, driven by a tissue-specific promoter (here, *ceh-60p*), is active. Heat shock promoters are used in non-heatshocked conditions to maintain a low and steady state of expression.

## 2.3 DamID: Sampling and library preparation

DamID sampling and library preparation were carried out essentially as described in [33]. Strains carrying either the *gfp*::*dam* (LSC1722) or *dam*::*ceh-60* (LSC1724) transgene were grown under standard conditions. Standard hypochlorite treatment was used to harvest eggs, which were allowed to hatch overnight in M9 buffer. For growing animals for DamID library preparation, four 100 mm NGM plates seeded with Dam-negative *E. coli* GM119 were used. Dam-negative bacteria do not show GATC methylation, which occurs naturally in most other *E. coli* strains and would otherwise contaminate *C. elegans* DamID libraries with bacterial GATC methylation. Young adult or L2 animals were rinsed off plates and 30 µL aliquots were snap-frozen in liquid nitrogen. Genomic DNA was extracted and purified using the Qiagen DNeasy Blood and Tissue kit according to the manufacturer's instructions. 200 ng of each genomic DNA sample was digested by incubation with 10 units of DpnI in 10 µL for 6 hours at 37˚C, followed by inactivation at 80˚C for 20 minutes. DpnI-digested DNA was incubated overnight at 16˚C with 2 µL ligation buffer, 1 µL Roche T4 DNA ligase and 0.8 µL of double-stranded adaptors (AdRt: CTAATACGACTCACTATAGGGCAGCGTGGTCGCGGCCGAGGA, 50 µM, AdRb: TCCTCGGCCG, 50 µM, mixed separately, heated to 95˚C and cooled to room-

temperature before addition to ligation mix) in a total volume of 20 μL. Ligase was inactivated by incubation for 10 minutes at 65˚C. DpnI-digested adaptor-ligated DNA fragments were purified with AgenCourt AMPure XP® beads and a magnetic particle concentrator, after which the sample was digested with DpnII (NEB) by incubation for 1 hour at 37˚C, followed by inactivation at 80˚C for 20 minutes and another purification step with AgenCourt AMPure XP® beads. DpnII-digested DNA was then amplified using Taq polymerase and 1.25 μL of 50 μM Adr primer (NNNNGGTCGCGGCCGAGGATC). Amplified adaptor-ligated DNA was analyzed by agarose gel electrophoresis and size-selected for amplicons between 400–1200 bp, the expected size of naturally-occurring GATC-flanked reads, using AgenCourt AMPure XP® beads and a magnetic particle concentrator according to the manufacturer's instructions. Using a 0.3 bead-to-sample ratio and selecting the supernatant, amplicons larger than 1200 bp are discarded. Next, using a 0.8 beads-to-sample ratio and selecting the bead-bound DNA, amplicons smaller than 400 bp are discarded. The DNA pool is separated by size through gel electrophoresis to confirm the presence of DNA only in the expected 400–1200 bp range. Adaptor ligated DNA is enriched by PCR and prepared for sequencing using the NEB-Next Singleplex oligos for Illumina® by adding 25 μL of NEBNext Q5 Hot Start HiFi PCR Master Mix to 5 μL Index Primer and 5 μL Universal PCR primer to 15 μL of Adaptor-ligated DNA fragments, followed by 8 cycles of PCR amplification as per the manufacturer's instructions. Finally, the sequencing libraries are purified with AgenCourt AMPure XP® beads and size of the amplicons is confirmed by agarose gel electrophoresis and Experion$^{TM}$ automated electrophoresis system (Bio-Rad) before sending out the libraries for Illumina sequencing at EMBL GeneCore (Heidelberg, Germany).

## 2.4 DamID: Data analysis

Next-generation sequencing data generated from DamID DNA samples was processed using a variant of the DamIDSeq *R* pipeline optimized for mapping DamID reads to gene regions termed GeneDamIDseq [34], freely available as *R* module. The GeneDamIDSeq module takes as input a text file describing the nature and path of (compressed) fastq files which contain the raw sequencing data for all conditions. Reads that do not contain the adapter sequence followed by the GATC motif are discarded, while the adapter sequence is trimmed from the remaining reads. Trimmed reads are mapped to the *C. elegans* reference genome BSgenome. Celegans.UCSC.ce11, obtained from WormBase, using Bowtie. Read counts for each binned genomic region are summed per sample and normalized for total read number per sample. In GeneDamIDSeq, the binned regions correspond to genes on the *C. elegans* genome.

The resulting output contains read numbers for each bin (*i.e.* each *C. elegans* gene) for each of three replicates for experimental (*i.e. dam*::*ceh-60*) and control (*gfp*::*dam*) conditions. First, genes were selected where the *dam*::*ceh-60* normalized read number in each single lane was higher than the read number in each single *gfp*::*dam* lane and fold-change of average read number of *dam*::*ceh-60* over *gfp*::*dam* was higher than 1.7. Correlation analysis on mapped GATC reads was performed by calculating Spearman's correlation coefficient for all comparisons within young adult or L2 datasets, after removing GATC sites with 0 or 1 read, leaving ~19,000 reads out of ~27,000 per sample.

Corresponding gene names for all targets were obtained using the BioMart Data Mining Tool on WormBase ParaSite. Gene Ontology analysis was carried out using PANTHER 15's statistical overrepresentation test using default settings for biological processes. Gene target lists of L2 and young adult animals were compared for significant overlap using a chi-square test. Single-cell RNA-seq data from [35] was mined for expression profiles of CEH-60 DamID gene targets in order to obtain expression values in each tissue for genes present in both L2

and young adult datasets, L2 only, and young adult only. For each list, the total transcripts per million value was calculated per tissue. DamID and ChIP-seq peaks were tested for overlap using ChIP-seq NarrowPeaks lists for CEH-60 or UNC-62 [9], and 3kb promoter regions upstream of CEH-60 targets identified by GeneDamIDseq. CEH-60 DamID *vs* CEH-60 ChIP-seq targets, and CEH-60 DamID *vs* UNC-62 ChIP-seq were compared for significant overlap using a chi-square test. Raw and processed DamID data was deposited to ArrayExpress (accession E-MTAB-9539).

## 2.5 Microscopy

For characterization of AWC neuron morphology, synchronized young adult animals carrying the *odr-1::rfp* transgene were anesthetized with 1 mM tetramisole and visualized using a confocal FluoView1000 microscope (Olympus, Japan). Z-stack images of the head region of each animal were converted into maximum intensity projections using Fiji [36].

**2.6 Chemotaxis assay.** Butanone chemotaxis assays were performed essentially as described in [37] for naive sensing conditions. Synchronized animals were grown at 20˚C on *E. coli* OP50 until the young adult stage, after which they were washed off the plates with M9 buffer and collected in 15 mL conical tubes. Worms were allowed to settle without centrifugation to minimize sampling stress and they were washed twice with M9 buffer to remove residual bacteria. Assay plates were prepared by spotting 1μL of 1M $NaN_3$ on both sides of the plate and additionally adding 1μL of 95% EtOH to one spot and 1μL of 10% butanone in a previously prepared 95% EtOH solution to the other spot. Approximately 100 animals per replicate were spotted at the origin. Chemotaxis assay plates were incubated for 1 hour at room temperature, after which the number or worms at the origin, EtOH-spot, butanone-spot and the total number of worms on the plate were counted. The chemotaxis index (CI) was calculated as

$$CI = \frac{n_{butanone} - n_{EtOH}}{n_{total} - n_{origin}}$$

Conditions were compared using a one-way ANOVA with Dunnett's multiple comparison post-hoc test.

## 2.7 Pharyngeal pumping assays

Pharyngeal pumping assays were performed essentially as described in [38]. Non-starved day 1 adult animals were filmed on NGM plates with *E. coli* OP50 for three times 15 seconds using a Leica M165 FC microscope outfitted with an MU035 AmScope microscope eyepiece camera. Recordings were played back at one fourth of normal speed to quantify pumping rate. Six animals were imaged per condition for three periods of 15 seconds each. Pumping rates were averaged per animal. Pumping rates for all conditions were compared using a one-way ANOVA with Dunnett's multiple comparison post-hoc test.

Isthmus peristalsis rate coinciding with movement of food from the corpus of the pharynx to the terminal bulb was measured using dsRed-expressing OP50 bacteria as a food source, as described in Van Sinay et al. [39] with some adaptations. Well-fed young-adult animals were transferred to a 2% agarose pad containing a 2 μL drop of OP50-dsRed *E. coli* bacteria, allowed to air dry and covered with a coverslip. Animals were imaged on a Zeiss Axio Observer Z1 equipped with a hEGFP/HcRed filter using only the 555 mm light source. MetaMorph® software was used for image acquisition. Image analysis was performed in Fiji [36]. Isthmus peristalsis rates were quantified during 100 seconds for at least six animals per condition. Frames during which the animal was out of frame or out of focus were censored. Rates were compared using a one-way ANOVA with Dunnett's multiple comparison post-hoc test.

## 3. Results and discussion

### 3.1 DamID reveals putative gene targets of CEH-60

To identify the gene targets of CEH-60, we performed DamID on L2 larvae and young adult animals carrying the *dam*::*ceh-60* transgene, using non-specific *gfp*::*dam* animals as controls. Highly reproducible results were obtained from three biological replicas as shown by Spearman's correlation analysis (S1 Fig). High correlation values are observed within *dam*::*ceh-60* samples ($\geq$0.67 in YA, $\geq$0.65 in L2). Correlation between *gfp*::*dam* and *dam*::*ceh-60* is also high ($\geq$0.52 in YA, $\geq$0.51 in L2). This is not surprising, as open chromatin is more accessible than dense chromatin to both proteins: transcription factor fusions, such as Dam::CEH-60, and diffusible GFP::Dam [40, 41]. We assigned loci as putative CEH-60 targets when the normalized *dam*::*ceh-60* read number in each single replica was higher than the normalized read number in each single *gfp*::*dam* lane, and the fold-change of average read number of *dam*::*ceh-60* over *gfp*::*dam* was higher than 1.7, resulting in 872 candidate gene targets in young adult samples (S1 Table).

Gene Ontology (GO) analysis of these 872 candidates reveals several statistically overrepresented biological processes, most prominent among which are categories related to muscle, embryo and neuron development (Fig 1 and S3 Table). Among the GO terms, most contain 20–40 gene targets, representing only a small set of the 872 hits, indicating that CEH-60 likely does not target a single pathway or process, but instead regulates many processes or pathways, possibly in subtle ways.

Looking at GO terms in more detail, CEH-60 target loci identified by DamID contain multiple genes regulating neuron projection development specifically, or neuronal development more generally. These include the FEZ1 ortholog *unc-76*, important for axon fasciculation and extension [42], *zag-1*, a Zn-finger homeobox transcription factor gene regulating axon development and neuronal differentiation [43], *unc-130*, coding for a Forkhead transcription factor involved in axon extension and guidance [44], the kinase gene *sad-1*, which regulates presynaptic vesicle clustering and axon termination [45], the cell-adhesion molecule gene *sax-7*, involved in neuronal development [46], the serine/threonine kinase gene *sax-1*, involved in the

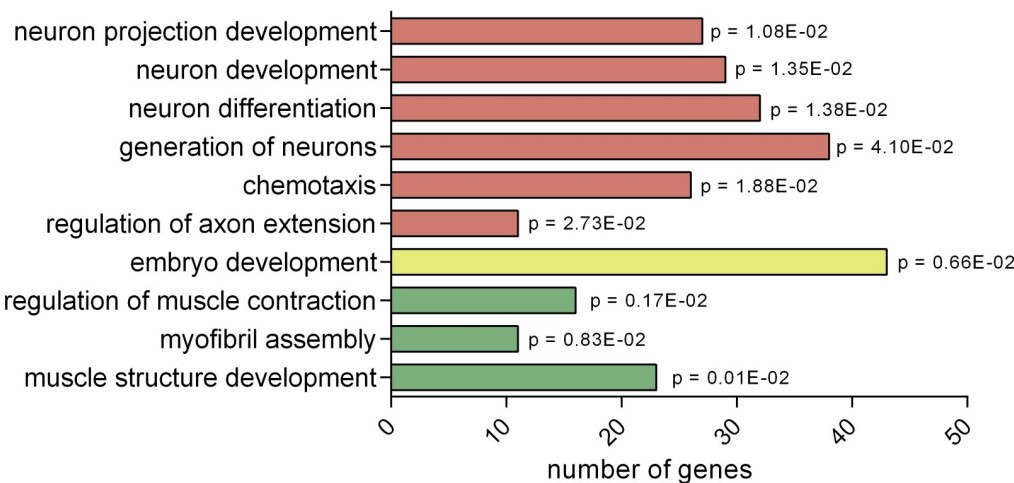

**Fig 1. Gene ontology analysis of 872 DamID gene targets in young adult animals.** Non-exhaustive list of biological processes overrepresented in CEH-60 DamID targets. False discovery rate corrected p values are shown for the specified biological process. GO terms were grouped thematically (neuron/embryo/muscle development) by color. Gene ontology analysis was carried out with PANTHER 15 using statistical overrepresentation test for biological processes (complete). A complete list of overrepresented GO terms can be found in S3 Table.

development of axons of sensory neurons [47], the contactin gene *rig-6*, which mediates neuronal cell migration [48], the synaptic guidepost protein coding gene *syg-2*, important for synaptic specificity [49], the heterochronic regulator of cell fate decision during larval development *lin-14* [50], the homeobox transcription factor gene *ceh-14*, which is important for development of thermosensory neurons [51] and the kinesin motor protein encoding gene *vab-8*, which functions in posteriorly directed neuron migration [52]. The presence of this large group of drivers of neuronal development, most of them involved in axon extension, suggests a role for CEH-60 in the development of *ceh-60* expressing AWC neurons and their related sensory function.

The other large group of candidate genes identified by gene ontology analysis is involved in muscle development and contraction. These include the troponin T1 orthologs *mup-2*, *tnt-2* and *T22E5.6*. These three genes form part of the troponin complex which enables $Ca^{2+}$-dependent muscle contraction and in *C. elegans* is also important for muscle development and axon fasciculation [47, 53], the myosin light chain-encoding genes *mlc-1*, *mlc-2*, *mlc-3* and *mlc-4* [54], the myosin heavy chain genes *myo-3* [55] and *unc-54* [56], the myosin light-chain kinase-like *unc-22* gene [57], the Obscurin-coding gene *unc-89*, a kinase important for muscle cell architecture [58, 59], the tropomyosin ortholog *lev-11*, and the paramyosin-coding gene *unc-15*, a major structural component of invertebrate muscles [60]. Many of the targets in this category code for large structural proteins or kinases in muscles, which suggests a possible role for CEH-60 in muscle structure. In support of this, the *ceh-60* paralogs *ceh-20* and *ceh-40*, and CEH-60's *in vivo* interaction partner UNC-62 have all been implicated in muscle patterning, vulval muscle development and post-embryonic muscle cell differentiation in *C. elegans* [2, 4–6]. Vertebrate *Pbx* genes, orthologs of *ceh-20*, *-40* and *-60*, are also known to be involved in muscle development [20–24].

While *ceh-60* is not expressed in the body wall muscle of *C. elegans*, which comprises the largest muscle system in the animal, *ceh-60* is distinctly expressed in the smaller pharyngeal muscle system [10], indicating a possible role for *ceh-60* in this tissue. As "regulation of muscle contraction" is one of the most overrepresented biological processes among CEH-60 DamID targets, we speculate that CEH-60 could be involved in contraction of the pharynx, which is indispensable for feeding [61]. Indeed, for 9 out of the 23 genes classified under "muscle structure development", the largest muscle-related biological process in the GO analysis (Fig 1), there is evidence for expression in pharyngeal muscle: *unc-89*, *unc-22*, *unc-15*, *emb-9*, *dyb-1*, *alp-1*, *unc-96*, *mlc-1* and *unc-27* [54, 62–69]. These observations indicate that *ceh-60*'s function in the muscle system could be related to the pharynx. Alternatively, CEH-60's action could involve the body wall muscles, but expression in these tissues may be low or time-specific.

Apart from muscle structure and neuronal development, embryonic development is also overrepresented among gene targets. While we did not study the function of CEH-60 in embryonic development in detail, a recent system-level screen of early embryonic development showed that *ceh-60* knockdown causes disorganized cell division during two early cleavage points of embryonic development (P3 and AB8) and sporadic misarrangement prior to division [70]. Curiously, *ceh-20* and *unc-62* knockdown were also tested in the high-throughput study by Guan et al. [70], but without apparent defects. This could indicate that the role of *ceh-60* in embryonic development may be mechanistically distinct from its postembryonic function, although no concrete evidence is present for this hypothesis.

Curiously, *unc-62* itself is also found among the candidate gene targets of CEH-60. UNC-62 is an intestinal interaction partner of CEH-60 important for lipid transportation [9, 10]. The possibility of genetic cross-regulation by direct binding of CEH-60/PBX to the UNC-62/MEIS-encoding DNA is interesting, and suggestive of a feedback mechanism for this conserved regulatory complex. One group of genes that is notably absent among the DamID gene targets however, encompasses those coding for vitellogenins, precursors to yolk proteins, discussed below.

## 3.2 Developmental expression shift of CEH-60 targets

We performed DamID on both young adult animals and L2 larvae in order to get a more dynamic view of how CEH-60 gene targets may change when its expression pattern shifts with respect to development, *i.e.* from limited to the neurons and PM6 during early larval stages to also including intestinal cells in L4 and adult stages [10]. Applying the same approach as for young adult animals, we found 587 candidate CEH-60 targets in L2 DNA (S2 Table). Of these, 43% (252/587 targets, p < 2.2E-16) overlap with those found in young adults. The overlapping genes include muscle proteins *mlc-3*, *myo-3*, *tnt-2*, *unc-15*, *unc-54* and *T22E5.10* and the neuronal development regulators *lin-14*, *unc-130* and *syg-2*, in addition to the vitellogenesis regulators *unc-62* and *lrp-2*. The observation that many genes are also "lost" from the L2 to the young adult stage suggests that early stage-specific marks placed by Dam::CEH-60 may be masked during development by an accumulation of methylation in open chromatin by the continuous activity of GFP::Dam in non-dividing cells. Indeed, we found no significant difference in signal intensity between overlapping and non-overlapping ($<\log_2 FC_{overlap}> = 1.36$, $<\log_2 FC_{non\text{-}overlap}> = 1.51$, $p = 0.29$) genes, indicating that non-overlapping genes are not simply noise in the L2 dataset that was not filtered out.

We compared the expression sites of L2 and young adult DamID gene targets to gain insight into possible shifts in binding preferences of CEH-60 upon reaching adulthood. For example, an increase in primarily intestinal genes in young adult animals (compared to larval animals) could reflect the activation of *ceh-60*'s intestinal expression from the L4 stage onward [9, 10]. To this end, we queried three lists of CEH-60 gene targets against publicly available single-cell RNA-seq data from L2 larvae [35]: (1) those overlapping between L2 and young adult, (2) those occurring only in L2 and (3) those occurring only in young adults. Because no peer-reviewed, whole-animal single-cell RNA-seq data are available that would allow a similar analysis based on expression patterns of young adults, we were limited to assigning genes—also the young adult ones—to larval expression patterns.

In presumed CEH-60 target genes that overlap between L2 and young adult datasets, body wall muscle expression is most enriched: body wall muscles represent 61% of all CEH-60 target gene expression, while all other tissues represent only 5–9% each. Of all body wall muscle transcripts measured in the Cao et al. dataset [35], $\frac{179,555}{1,000,000} = 18\%$ are represented by CEH-60 DamID targets found in both L2 and young adults, while this value ranges from 1–2.5% for other tissues.

In genes identified as possible CEH-60 targets in the L2, but not in the young adult stage, no clear tissue preference can be observed: as opposed to what is seen in the shared gene group, here, body wall muscles account for a share of only 19% of target gene expression, which is similar to the 12–17% weight of nearly all other tissues: gonad, hypodermis, neurons, pharynx and glia. Intestinal expression is least abundant, accounting for only 9%.

In genes present as putative CEH-60 targets only in young adults, but not in L2 larvae, body wall muscles again take the lead in tissue weight, claiming 39% of target gene expression, while neurons and pharynx represent 15% and 12% respectively. Based on *ceh-60*'s known temporal change in expression pattern [9, 10], the limited weight of the intestine in this young-adult-only list may initially seem counter-intuitive. We cannot emphasize explicitly enough, however, that assignment of tissue-resolved expression is based on RNA-seq data from L2 larvae [35]. CEH-60 gene targets may be expressed in different tissues throughout life, just like *ceh-60* itself, and targets may for example be expressed in the intestine in young adults, but not in L2 larvae. Thus, our expression analysis should be used as an indication of which categories of genes may be targeted by CEH-60, until future single-cell resolved sequencing data of adult animals will become available that will allow to fully account for life-stage-specific changes. In

all three lists, body wall muscles are the most highly represented, corresponding to the previously described GO-proposed roles of CEH-60 in muscle function. In the overlapping gene list, body wall muscle expression is highest (61 *vs* 19% and 39%), which could be indicative of a constitutive 'core function' of CEH-60 in muscles throughout postembryonic life. Neurons represent less expression in overlapping genes than in L2 or young adult-specific genes, which may indicate stage-specific neuronal roles for CEH-60. The intestine represents only 6–10% of total expression in all three datasets, although expression is highest for young-adult only genes, which may reflect the reported increase in intestinal CEH-60 from the L4 stage onwards (*cf* above, [9, 10]).

## 3.3 DamID and ChIP-seq are complementary tools for identifying gene targets and revealing new transcription factor functions

When comparing our DamID results with those based on an available ChIP-seq dataset [9], we find that 301 of 872 DamID-based CEH-60 target genes have one or more corresponding ChIP peaks (35%, p < 2.2E-16). Our result of 35% corresponds well with percentages of 32–49% reported for similar comparisons in literature [30, 71]. While this overlap is convincing, it does show that many DamID targets are not found by ChIP-seq and vice versa. This indicates that DamID and ChIP-seq approaches could be complementary when searching for gene targets of transcription factors. Possibly, sterical considerations limit the interactions revealed by either technique: a POI::Dam fusion protein may have different access to the DNA than a POI::GFP protein pulled down with immunoprecipitation. Additionally, the relative accessibility of different tissues might differ between the two techniques.

Among the overlapping targets, many belong to the broad functional categories delineated above, including muscle proteins (*lev-11, mlc-1, -2 and -4, mup-2, T22E5.10, tnt-2, unc-54, unc-89*) and regulators of neuronal development (*lin-14, rig-6, sad-1, sdn-1, unc-130, unc-73, vab-8*). On the other hand, one notable set of targets lacking in our DamID-based data are a group of intestinally-expressed genes, the *vit* genes.

*vit* genes code for vitellogenin proteins, and are lacking from our gene target list in both young adult and L2 animals, even though they are obvious candidates for direct regulation by intestinal CEH-60 in young adults [9–11]. A recent study of CEH-60 gene targets employing a different approach, did find that CEH-60 directly interacts with the promoters of *vit* genes in adult animals [9]. In line with this, CEH-60's known interaction partner UNC-62 is also known to bind to the VPE1 element in the *vit* gene promoters [72, 73]. One possibility for the peculiar absence of *vit* genes in our dataset, is that our DamID pipeline may have been more stringent than standard ChIP-seq analysis, as indicated by the 872 gene targets identified through DamID by us, and the 9010 targets identified through ChIP-seq [9]. Alternatively, our DamID sampling may not have been optimal for capturing binding of *vit* genes in the *C. elegans* intestine, as this only occurs from the L4 stage on. While we sampled during the young adult stage, methylation of the GATC sites in *vit* promoters may have required more time to become significantly enriched. This could be a second reason why we are not observing an increase in intestinal targets in young adult animals compared to L2 larvae (Table 2), in addition to the L2 origin of the tissue-specific RNA-seq data (*cf* above).

When comparing ChIP-seq data of CEH-60's known interaction partner UNC-62 [9] with DamID targets of CEH-60, we find that 82 out of 872 CEH-60 DamID targets correspond to one or more of the 2053 ChIP-seq peaks for UNC-62 (9.4%, p = 8.21E-13). This shows that there is a significant overlap in gene targets of CEH-60 and UNC-62, as would be expected from their conserved *in vivo* interaction. Of these 82 shared genes between CEH-60 DamID and UNC-62 ChIP-seq targets, 69 also occur in the 301 overlapping targets of CEH-60 DamID

**Table 2. Distribution of expression levels over tissues for CEH-60 gene targets identified by DamID present in both L2 and Young Adult (YA) datasets, L2 only or YA only, according to L2-stage single-cell RNA-seq analysis of [35].**

| Tissue | (1) L2-YA shared | | (2) L2 only | | (3) Young adult only | |
|---|---|---|---|---|---|---|
| | tpm | % | tpm | % | tpm | % |
| Body wall muscle | 179555 | 60.76 | 47782 | 18.80 | 167937 | 39.05 |
| Glia | 18978 | 6.42 | 36195 | 14.24 | 42751 | 9.94 |
| Gonad | 18425 | 6.24 | 41866 | 16.47 | 28815 | 6.70 |
| Hypodermis | 14812 | 5.01 | 38819 | 15.27 | 33500 | 7.90 |
| Intestine | 18744 | 6.34 | 22801 | 8.97 | 40098 | 9.32 |
| Neurons | 24033 | 8.13 | 36197 | 14.24 | 64603 | 15.02 |
| Pharynx | 20948 | 7.09 | 30502 | 12.00 | 52389 | 12.18 |

Body wall muscle expression is most abundant in all three gene target lists. "tpm" indicates transcripts per million, this is the number of transcripts represented by genes in the specified list, per one million total transcripts in that tissue (*i.e.* relative weight of the target gene list within the total transcriptome of the tissue). "%" indicates the percentage of total gene expression represented by the specified tissue in that dataset (*i.e.* tissue enrichment per list: each column adds up to 100%).

and CEH-60 ChIP-seq datasets (84.1%, p < 2.2E-16). Together, these data show that gene targets generated using DamID share a significant number of hits with gene targets generated using other methods for the same transcription factor or transcription factors that are—at least in some proven cases—part of the same complex, such as UNC-62.

Comparing our results to other DamID studies reveals that the number of putative gene targets identified by us, 872 for YA and 587 for L2 animals, is within the expected range [27, 29, 74, 75]. Still, false positives are definitely possible and gene targets identified by DamID alone should be regarded as putative. We here relied on DamID as a tool to build hypotheses on new functions of the transcription factor CEH-60, which can be subsequently studied using functional assays.

## 3.4 CEH-60 is not essential for normal AWC morphology or odor-sensing function

The transcription factor CEH-60 targets several genes important for neuronal development (Fig 1), and *ceh-60* is expressed throughout life in the AWC neurons [10]. To assess whether the morphology of these neurons depends on CEH-60's function, we crossed *ceh-60(lst466)* mutants with the AWC-specific *odr-1p::rfp* marker and observed the morphology of AWC neurons in both conditions (Fig 2). RFP signal can clearly be observed in the cytoplasm of both wild-type and *ceh-60* mutant AWC neurons, with sensory cilia being present and reaching the tip of the nose of the animal. Also, the dendrite connecting the left and right AWC neuron by following a semicircular path along the nerve ring, appears similar in both conditions.

We additionally tested whether despite normal overall morphological appearance, AWC function might be affected in *ceh-60* mutants. The AWC neurons are best-characterized as sensors of volatile odors such as butanone [13]. Under normal circumstances, *C. elegans* is attracted to butanone. We found that the chemotaxis index (CI, Fig 3A), which here quantifies a population's attraction to butanone, does not differ between wild-type animals and either of the tested *ceh-60* mutant alleles (Fig 3B), despite a proper response of the positive control (*che-3* mutants, deficient in a known regulator of volatile odorant sensing [13]). This indicates that the odorant-sensing apparatus, which includes the AWC neurons, is functional in absence of CEH-60, and taken together with the morphological data (Fig 2), likely intact.

As tested by these assays, CEH-60 does not appear to affect AWC morphology or function. However, besides the here presented gene target studies and *ceh-60*'s expression in AWC,

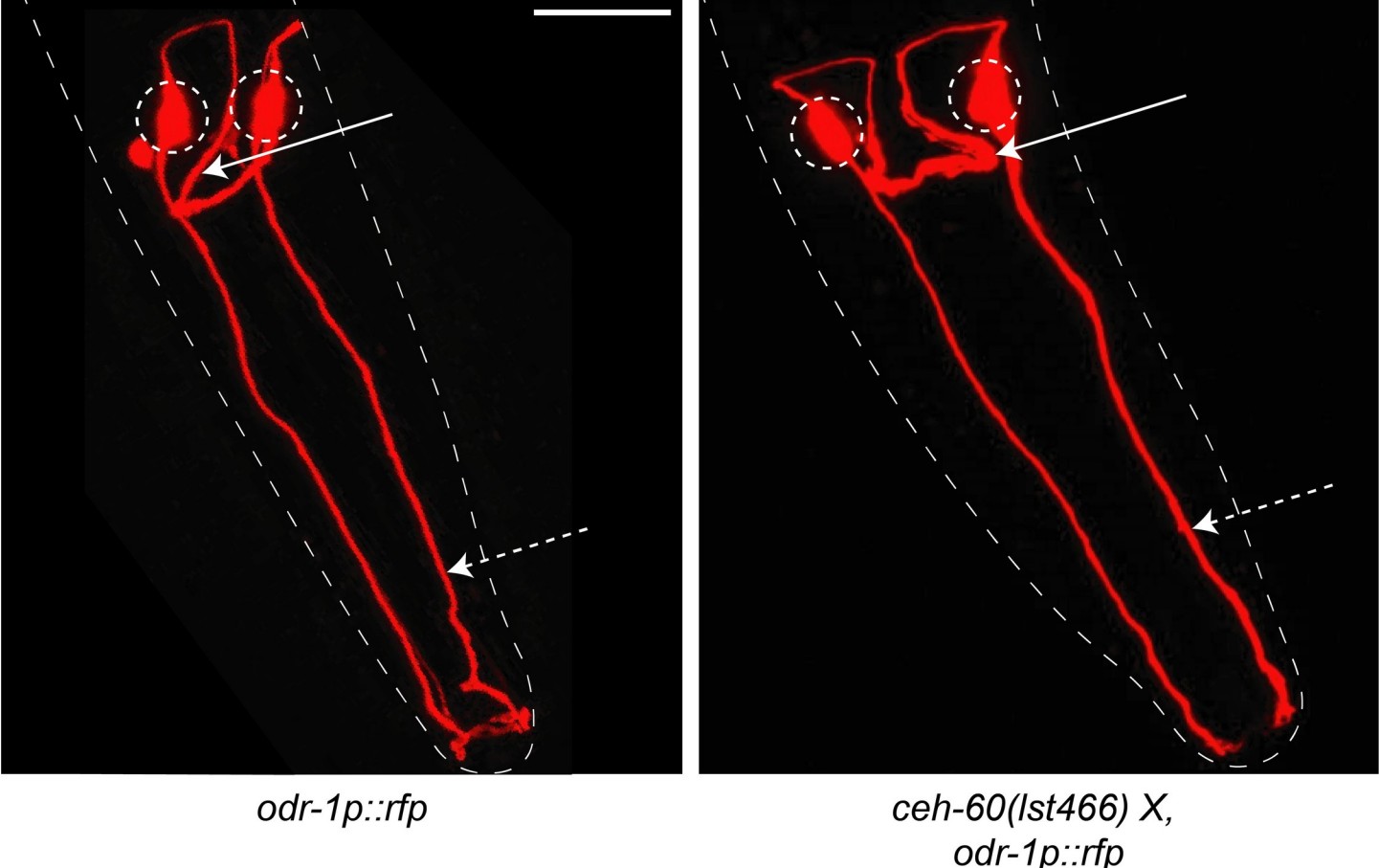

*odr-1p::rfp*                                    *ceh-60(lst466) X,*
                                                 *odr-1p::rfp*

**Fig 2. Morphology of AWC neurons in the head of control and *ceh-60(lst466)* mutant animals is similar.** All animals carry the AWC-specific *odr-1p::rfp* fluorescent marker. We found no clear difference in morphology of the AWC neurons between mutants and controls. In both cases, the neuron bodies (dashed circles) are clearly formed and extend their projections towards the sensory tip of the nose (dashed arrows) and the nerve ring (solid arrows). Outline of animals is shown as a dotted white line. Scale bar = 20 μm. Both animals are L4 larvae.

there are other reasons to believe that CEH-60 has a role to play in olfactory neurons. Previously, CEH-60 has been characterized as a modulator of fat mobilization through activation of vitellogenesis in the intestine [9, 10]. Olfactory sensing has often been linked to fat storage and metabolism, with a high-fat diet in mice causing decreased sense of smell and olfactory neuron activity [76], and human *anorexia nervosa* patients often experiencing increased olfactory sensing [77]. Recently, butanone sensing through AWC neurons in *C. elegans* was also found to influence fat storage and mobilization, likely signaling through the neuropeptide FLP-1 and its receptor NPR-4, the glucocorticoid-inducible kinase GSK-1 and DAF-16 in peripheral tissues [78]. Because CEH-60 does not appear to influence butanone sensing, we maintain that the method through which CEH-60 alters fat mobilization and storage, is through its documented regulation of vitellogenesis.

Future work may yet unveil CEH-60's cryptic role or function in olfactory AWC neurons, further motivated by *(1)* the recent discovery that PBX1, coding for *ceh-60*'s homolog in vertebrates, acts as a terminal selector for olfactory bulb neuron differentiation in mice [79], and *(2)* the knowledge that in *C. elegans*, olfactory learning depends on the odor-sensing (*ceh-60*-expressing) AWC neurons but also involves signaling from DBL-1/TGF-β [80], which is here identified as a putative gene target for CEH-60 (S1 Table). DBL-1 could indeed be the signaling

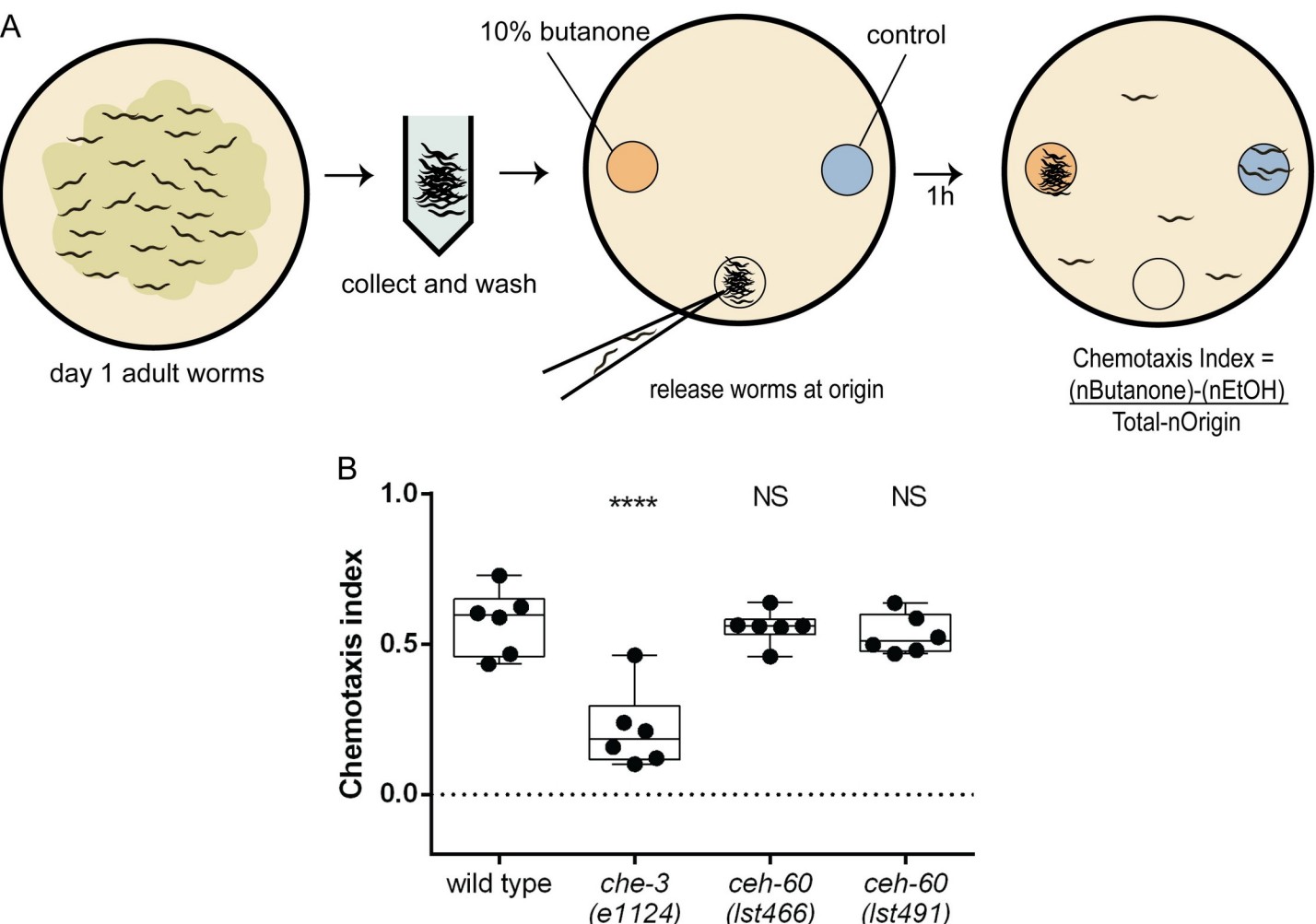

**Fig 3. *ceh-60* mutants have normal butanone sensing capacity.** (A) The butanone sensing assay experimental setup. Animals are collected, washed to remove residual bacteria and released at the origin of a chemotaxis plate, to which 10% butanone (orange) and control (blue) spots have been applied (details: methods). After 1 hour, the chemotaxis index is calculated as CI = ([(nButanone)-(nEtOH)]/[(Total-nOrigin)], with n the number of animals at these positions. (B) Wild-type animals are able to sense butanone, as evidenced by their positive CI. *che-3(e1124)* animals have a documented defect in sensing butanone, and their (significantly lower) CI is used as a positive control for the assay. Neither *ceh-60* mutant strain shows a significant difference in CI. **** p < 0.0001. NS, not significant. Each dot represents an assay of a biologically independent population, N = 6.

molecule linking several CEH-60-related phenotypes: DBL-1 regulates vitellogenesis [73], is needed for building of an impermeable epicuticle [81], functions as an olfactory learning cue in the neurons [73, 80, 81], and is a gene target of CEH-60 as identified by DamID (this study). So far however, the exact role of CEH-60 in AWC neurons remains undiscovered. Because simple morphological and functional studies here performed showed no distinct phenotype, it is possible that the function of CEH-60 in AWC neurons is subtle, redundant with other (transcription) factors, or unrelated to neuron morphology or easily quantifiable behavior.

### 3.5 CEH-60 is not essential for on-food pharyngeal pumping

The *C. elegans* hermaphrodite muscle system consists of body wall, pharyngeal, anal sphincter, anal depressor, vulval and uterine muscles and a contractile gonadal sheath [82]. Because many CEH-60 targets are involved in muscle contraction and are expressed in the pharyngeal

muscle, and because *ceh-60* itself is expressed in the PM6 pharyngeal muscle cells, we decided to investigate pharyngeal pumping in *ceh-60* mutants by quantifying the pumping rate in the presence of food of adult animals. While *eat-2* mutants, which have a well-documented defect in pharyngeal pumping rate [38] indeed show a decreased average pumping rate compared to wild types, *ceh-60* mutants do not (Fig 4A). Additionally, we quantified isthmus peristalsis, a contraction of pharynx muscles that carries food from the corpus of the pharynx to the terminal bulb, using fluorescent *E. coli* OP50 bacteria. We observed no difference between wild type and *ceh-60* mutant animals (Fig 4B). Thus, the function of *ceh-60* in muscle tissue is still to be determined.

DamID results (S1 and S2 Tables), *ceh-60* expression patterns [10] and *Pbx* functions in vertebrate muscle development [20–24] all suggest a role for CEH-60 in muscle structure, specifically in the pharynx, yet both pharyngeal contractions measured were not measurably affected in *ceh-60* mutants (Fig 4). CEH-60 function in the pharynx may not be apparent when studying on-food behavior, and may require more demanding conditions such as the absence of food, which normally leads to serotonin-dependent enhanced pumping [83].

While our dataset of CEH-60 targets also contains several genes known to be expressed in body wall muscle, CEH-60 itself is absent from this tissue [10, 35]. Because CEH-60 can also act as a repressor of transcription [9], these data could indicate that in the pharynx, CEH-60 actively represses transcription of certain body wall muscle genes. It would be interesting to explore this further in future research.

CEH-60's function in the pharynx may not be related to muscle structure directly. Recently, it has been shown that the PM6 pharyngeal muscle cells, in which *ceh-60* is abundantly expressed and which surround the pharynx grinder, transdifferentiate into secretory cells during lethargus, possibly aiding in the construction of a new grinder [84]. The grinder is an extracellular matrix, constructed during each larval transition phase, coinciding with earlier-reported cyclic expression of *ceh-60* [11, 85]. CEH-60 may act in PM6 cells to aid in this cyclic transdifferentiation from muscle cells to secretory cells, or in re-establishment of muscle nature once the extracellular matrix of the grinder is built.

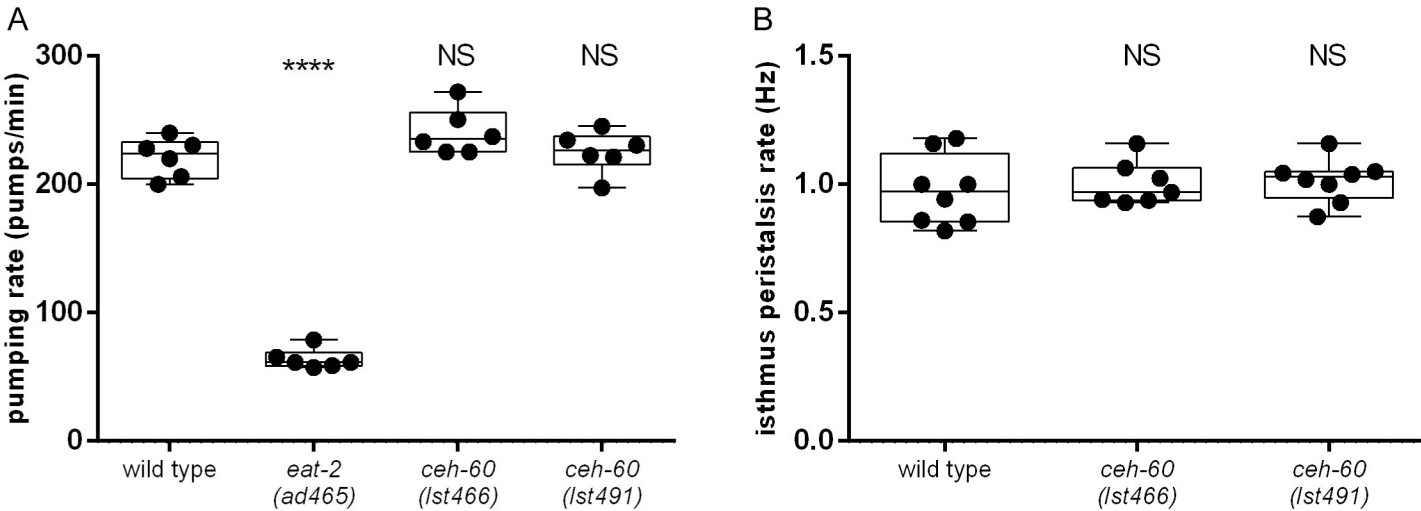

**Fig 4. *ceh-60* mutants have normal pharyngeal functions.** (A) Wild-type animals and *ceh-60* mutant animals have similar pumping rates on food, while *eat-2* mutants pump significantly slower. Each dot represents one animal (N), imaged thrice for 15 seconds. Pumping rates were averaged per animal. **** p < 0.0001. NS = not significant. N = 6. (B) Wild-type animals and *ceh-60* mutants have similar isthmus peristalsis or "gulp" rates. NS = not significant. N ≥ 6.

# 4. Conclusions

Expression patterns and functional information from homologs and paralogs all point towards the possibility for new roles of the transcription factor CEH-60/PBX in *C. elegans*. Through DamID, we identified gene targets of CEH-60 and hypothesized on its function in neuron development and muscle structure. Specifically, we tested morphology and function of sensory AWC neurons and pharyngeal muscle. While we did not find evidence for CEH-60-related function in the observed assays, our phenotypic analysis was limited to simple behavioral assays and morphological characterization of neurons. This means that roles for CEH-60 in neurons and muscle are still possible, although they may be subtle or hard to uncover because of genetic redundancy. In addition, we compared our DamID results to available ChIP-seq data and conclude that while there is a common core of genes identified, both techniques also identify unique targets.

# Supporting information

**S1 Fig. Spearman's rank-order correlation coefficients for *dam::ceh-60* and *gfp::dam* samples.** In both young adults (A) and L2 larvae (B), high correlation values are observed within *dam::ceh-60* samples ($\geq$0.67 in YA, $\geq$0.65 in L2). Correlation between *gfp::dam* and *dam::ceh-60* is also high ($\geq$0.52 in YA, $\geq$0.51 in L2).
(DOCX)

**S1 Table. 872 candidate gene targets of CEH-60 identified in young adult animals through DamID.** Start and stop represent the genomic positions on the specified chromosome (chr) of the specified open reading frame. Candidate gene targets are sorted by genomic position (*i.e.* chromosome number and start position). Log$_2$FC is calculated as the log$_2$ value of the ratio of average number of *dam::ceh-60* reads over average number of *gfp::dam* reads.
(DOCX)

**S2 Table. 587 candidate gene targets of CEH-60 identified in L2 animals through DamID.** Start and stop represent the genomic positions on the specified chromosome of the specified open reading frame. Candidate gene targets are sorted by genomic position (*i.e.* chromosome number and start position). Log$_2$FC is calculated as the log$_2$ value of the ratio of average number of *dam::ceh-60* reads over average number of *gfp::dam* reads.
(DOCX)

**S3 Table. Complete gene ontology analysis of 872 DamID gene targets in young adult animals.** Complete list of biological processes overrepresented in CEH-60 DamID targets. Observed and expected columns represent the number of genes associated with the specified ontology term and the expected number in a random collection of genes for the *C. elegans* genome. Fold enrichment represents the number of times the specified process is overrepresented in the dataset. FDR p value represents the false discovery rate corrected p value for the specified biological process. GO terms were sorted by FDR p value. Gene ontology analysis was carried out with PANTHER 15 using a statistical overrepresentation test for biological processes (complete).
(DOCX)

# Acknowledgments

OP50-dsRed *E. coli* bacteria were a gift from Dr. Andre Brown (Imperial College London, UK). Some strains were provided by the *Caenorhabditis* genetics center (University of Minnesota, US).

## Author Contributions

**Conceptualization:** Pieter Van de Walle, Peter Askjaer, Liesbet Temmerman.

**Data curation:** Pieter Van de Walle, Peter Askjaer.

**Formal analysis:** Pieter Van de Walle, Celia Muñoz-Jiménez, Peter Askjaer, Liesbet Temmerman.

**Funding acquisition:** Pieter Van de Walle, Peter Askjaer, Liliane Schoofs, Liesbet Temmerman.

**Investigation:** Pieter Van de Walle, Celia Muñoz-Jiménez, Peter Askjaer, Liesbet Temmerman.

**Methodology:** Pieter Van de Walle, Celia Muñoz-Jiménez, Peter Askjaer.

**Project administration:** Liliane Schoofs, Liesbet Temmerman.

**Resources:** Peter Askjaer, Liesbet Temmerman.

**Software:** Peter Askjaer.

**Supervision:** Peter Askjaer, Liliane Schoofs, Liesbet Temmerman.

**Visualization:** Pieter Van de Walle, Peter Askjaer.

**Writing – original draft:** Pieter Van de Walle, Celia Muñoz-Jiménez, Peter Askjaer, Liesbet Temmerman.

**Writing – review & editing:** Pieter Van de Walle, Celia Muñoz-Jiménez, Peter Askjaer, Liliane Schoofs, Liesbet Temmerman.

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
