## [Decision Letter · Decision Letter 0]

18 Sep 2020

PONE-D-20-27700

DamID identifies targets of CEH-60/PBX that are associated with neuron development and muscle structure in Caenorhabditis elegans

PLOS ONE

Dear Dr. Van de Walle,

Thank you for submitting your manuscript to PLOS ONE. After careful consideration, we feel that it has merit but does not fully meet PLOS ONE’s publication criteria as it currently stands. Therefore, we invite you to submit a revised version of the manuscript that addresses the points raised during the review process.

In particular it would be important to address in more details the conflicting expression data for the putative targets of CEH-60.

We look forward to receiving your revised manuscript.

Kind regards,

Denis Dupuy, Ph.D.

Academic Editor

PLOS ONE

Journal Requirements:

Reviewers' comments:

Reviewer's Responses to Questions

**Comments to the Author**

1. Is the manuscript technically sound, and do the data support the conclusions?

Reviewer #1: Yes

Reviewer #2: Partly

2. Has the statistical analysis been performed appropriately and rigorously? 

Reviewer #1: Yes

Reviewer #2: I Don't Know

3. Have the authors made all data underlying the findings in their manuscript fully available?

Reviewer #1: Yes

Reviewer #2: Yes

4. Is the manuscript presented in an intelligible fashion and written in standard English?

Reviewer #1: Yes

Reviewer #2: Yes

5. Review Comments to the Author

Reviewer #1: Van de Walle et al. investigate the role of the CEH-60/PBX transcription factor in neuron and muscle development in C. elegans using a molecular and a phenotypic approach. While the investigation does not identify any roles for CEH-60 in neuron or muscle development/function, the logic and rationale for the investigation are solid and the negative results are likely to be of interest to various sub-disciplines. For example, ceh-60 is expressed in some muscles and AWC sensory neurons and PBX-family TFs have known roles in muscle and neuron development across species, including paralogs in worm.

This work does not rule out a role for CEH-60 in nueorn and muscle, and these caveats are made known, although should be summarized in the conclusion (see below).

The authors use a DamID approach to identify potential CEH-60 binding sites in the genome, and through extrapolation, target genes that are regulated. Since this study is done at different time points, the data suggest possible changes in target gene regulation. While DamID is not the newest technology that could be used for this approach, it is still appropriate especially in metazoa. The DamID experiments appear to have been done with great care, appropriate controls, and in triplicate with good data correlation across repeats. This data set, which the authors indicate will be made available online, should be a useful resource for the community.

I commend the authors for a refreshingly well-written manuscript that was easy to read because of a strong flow of logic, good attention to detail, and adherence to standard genetic nomenclature. I only noted a few errors and places where clarification or citations are needed (see below).

Specific comments to address:

Line 63: Would be useful to also cite Choksi et al 2006 as a more recent and excellent example of DamID use in fly.

Lines 94 and 97: FLP_D5 is used with and without an underscore (with underscore in the table). Simply needs consistent use throughout.

Lines 215-216: There is a statement that GFP prefers open chromatin and this is used as a rationale for why CEH-60::Dam and GFP::Dam have a high correlation with each other. The statement that GFP prefers open chromatin would need a reference to support this. While it seems likely that GFP::Dam would lead to random methylation in open chromatin at a low frequency, it seems overstated to say it prefers open chromatin as GFP has no DNA binding affinity on its own. It also seems concerning that a negative control has significant correlation with the experiment suggesting that random DNA methylation is not too different than TF-driven methylation. This point should be addressed, perhaps by comparing this to past DamID results in other papers in fly/worm assuming similar results were obtained there for TF DamID experiments and GFP controls.

Lines 316-318: Similar to above, there is another claim about GFP::Dam continuously marking open chromatin, which can hide a signal. This should also be referenced citing past examples of this with other DamID studies.

The Table associated with the strains used and genotypes is listed as Table 3 yet is presented as the first table in the paper. Seems like it should be Table 1, given that do not see a table numbered as #1 and the next table to appear is #2.

Conclusion: A more thorough discussion of the caveats of the study are appropriate here such as the fact that CEH-60 may have roles in neuron and muscle that have not been uncovered here simply because of the limited scope of phenotypic analysis and possible genetic redundancy. Such information is alluded to in other places of the manuscript but a concise statement of the caveats in the conclusion seems appropriate.

Reviewer #2: 1. Its difficult to build a compelling story focused on the role of a gene product in particular cell types (AWCs and PM6) when the gene lacks a mutant phenotype in those cell types. The authors looked at two different ceh-60 mutants for phenotypes in AWC and PM6, but failed to detect functional deficits. One possible explanation raised by the authors is redundancy. I wonder whether phenotype would be detected if the authors built a dominant-negative type of construct whereby the DNA-binding domain was intact, but the transactivating domain lacked activity, thereby displacing redundant factors but preventing activity. Or engineering a hypermorphic state by replacing ceh-60’s transactivating domain with a ‘stronger’ one (whatever that means).

Based on the Cao et al data, I would not put much faith in ceh-60 expression in AWC (see below), so I would not focus a phenotypic search there. But, there is good expression of ceh-60 in the pharynx muscle. PM6 surrounds the grinder, and there are some fascinating biology that is going on there (see David Raizen’s 2020 PLoS One paper). Perhaps ceh-60 is involved in grinder formation, or the transdifferaction of surrounding cells, or re-establishing the muscle-nature of PM6 after it has transdifferentiated. If ceh-60 had an important role in grinder formation, I would expect a slow-growth phenotype, but no one has reported that, so maybe there is nothing there. The authors could look at the maceration of fluorescent bacteria for example to quantify grinder efficiency. Should the authors look at other large-scale datasets, they will find that ceh-60 oscillates in a time sequence in larval development that is consistent with a role in grinder formation or some other aspect of the larval development cycle.

2. The body wall muscle targets are very perplexing. If one examines the Cao et al 2017 data in detail, one will see that ceh-60 transcripts are detected at only very low levels (~40X less than in the intestine and ~100X less than in the pharynx muscle). Several of those genes are well-known to be exclusively expressed in non-pharynx muscle (unc-54, myo-3 etc). This, in combination with the lack of vit-gene targets detected, call into question the validity of the experiments (sorry). It's a head scratcher and some things don't add up.

On that note, the Cao et al data shows no expression of ceh-60 in ciliated neurons (which would include AWC) in L2s, which should raise some concern about reporter expression of ceh-60 in the AWC being a result of transgene artifact. (Sorry, but its not clear if the authors are making a claim that ceh-60 is expressed in the AWCs only in adults; if so, then the Cao data would not necessarily be relevant.

The authors might want to cross-reference their list of targets (as revealed by DamID or ChIP) with theose genes that are co-expressed with ceh-20 in the pharynx muscle. These would be high-value targets and worth further discussion.

Minor point: On line 280, it is not clear what the authors mean by writing, ‘9 are specifically expressed in pharyngeal muscle’. Certainly, they must not mean ‘exclusively expressed in pharynx muscle’, since those genes are known to be expressed in BWMs. Perhaps they mean that there is good evidence that they are also expressed in the pharynx. If so, they should clarify that sentence.

6. PLOS authors have the option to publish the peer review history of their article (what does this mean?). If published, this will include your full peer review and any attached files.

Reviewer #1: No

Reviewer #2: No

---

## [Author Response · Author response to Decision Letter 0]

4 Nov 2020

Reviewer 1:

Line 63: Would be useful to also cite Choksi et al 2006 as a more recent and excellent example of DamID use in fly.

This suggestion has been implemented in the revised manuscript on line 61. 

Lines 94 and 97: FLP_D5 is used with and without an underscore (with underscore in the table). Simply needs consistent use throughout.

FLP is now used throughout the revised manuscript for simplicity, as the addition “_D5” is not relevant in this manuscript.

Lines 215-216: There is a statement that GFP prefers open chromatin and this is used as a rationale for why CEH-60::Dam and GFP::Dam have a high correlation with each other. The statement that GFP prefers open chromatin would need a reference to support this. While it seems likely that GFP::Dam would lead to random methylation in open chromatin at a low frequency, it seems overstated to say it prefers open chromatin as GFP has no DNA binding affinity on its own. It also seems concerning that a negative control has significant correlation with the experiment suggesting that random DNA methylation is not too different than TF-driven methylation. This point should be addressed, perhaps by comparing this to past DamID results in other papers in fly/worm assuming similar results were obtained there for TF DamID experiments and GFP controls.

We apologize that the original text was unclear on this point and we have rephrased the text in the revised manuscript (lines 221-224): 

“This is not surprising, as open chromatin is more accessible than dense chromatin to both proteins: transcription factor fusions, such as Dam::CEH-60, and diffusible GFP::Dam (Sha et al. 2010; Sen et al. 2019)”. 

As discussed in the early DamID protocols, “methylation by dam is modulated by the local structure of chromatin: DNA in condensed chromatin, such as heterochromatin, is generally less accessible to dam than DNA in transcriptionally active, decondensed euchromatin” (Greil et al, Methods in Enzymology, Volume 410, 2006, Pages 342-359). Therefore, a diffusible Dam-only or Dam-GFP fusion is included to normalize for the different degrees of chromatin accessibility across the genome. Similar to our study, other C. elegans groups also use Dam-GFP rather than Dam-only as control (e.g. BMC Genomics. 2010 Aug 6;11:465. doi: 10.1186/1471-2164-11-465; Mol Syst Biol. 2010 Aug 10;6:399. doi: 10.1038/msb.2010.54.). The modified sentence includes a reference to a recent study in flies that used DamID to study chromatin accessibility and the interaction of Hunchback with target genes (https://doi.org/10.7554/eLife.44036). In line with our results and with the above, this study also finds a high correlation between Dam-Hunchback and Dam-only signal, both globally (Figure 6A-C), and at particular loci (Figure 6D-E).

Lines 316-318: Similar to above, there is another claim about GFP::Dam continuously marking open chromatin, which can hide a signal. This should also be referenced citing past examples of this with other DamID studies.

We are not aware of other studies that have analyzed non-dividing cells at two different developmental time points with basal or “leaky” expression of Dam fusion proteins. We have rephrased the text to more precisely reflect that this is an assumption based on our observations, rather than a fact (lines 319-322 in the revised manuscript): 

“The observation that many genes are also “lost” from the L2 to the young adult stage suggests that early stage-specific marks placed by Dam::CEH-60 may be masked during development by an accumulation of methylation in open chromatin by the continuous activity of GFP::Dam in non-dividing cells.”

The Table associated with the strains used and genotypes is listed as Table 3 yet is presented as the first table in the paper. Seems like it should be Table 1, given that do not see a table numbered as #1 and the next table to appear is #2.

Apologies for the oversight. Table 1 is now used for the strain table, both in table legend and revised manuscript text.

Conclusion: A more thorough discussion of the caveats of the study are appropriate here such as the fact that CEH-60 may have roles in neuron and muscle that have not been uncovered here simply because of the limited scope of phenotypic analysis and possible genetic redundancy. Such information is alluded to in other places of the manuscript but a concise statement of the caveats in the conclusion seems appropriate.

To implement this suggestion, the lines in the revised conclusion now read (lines 539-543 in the revised manuscript): 

“While we did not find evidence for CEH-60-related function in the observed assays, our phenotypic analysis was limited to simple behavioral assays and morphological characterization of neurons. This means that roles for CEH-60 in neurons and muscle are still possible, although they may be subtle or hard to uncover because of genetic redundancy.”

Reviewer 2: 

The authors looked at two different ceh-60 mutants for phenotypes in AWC and PM6, but failed to detect functional deficits. One possible explanation raised by the authors is redundancy. I wonder whether phenotype would be detected if the authors built a dominant-negative type of construct whereby the DNA-binding domain was intact, but the transactivating domain lacked activity, thereby displacing redundant factors but preventing activity. Or engineering a hypermorphic state by replacing ceh-60’s transactivating domain with a ‘stronger’ one (whatever that means).

While we can only agree that it would be very interesting to study the effects of modifying different CEH 60 domains, this is not yet possible because it first requires additional insight into its protein complex; information that is currently lacking. We already know that CEH-60 associates with at least two other transcription factors in spatiotemporally restricted ways (UNC-62 and PQM-1, Dowen et al. 2019 and Van de Walle et al. 2019), and that depending on unknown interactions, CEH-60 can switch between acting as a repressor vs activator of transcription. Because the dynamic nature of the transcription factor is complex and because its specific interactions are largely unknown, it is not yet possible to make strong hypotheses on expected outcomes of the alleles suggested by the reviewer. 

Based on the Cao et al data, I would not put much faith in ceh-60 expression in AWC (see below), so I would not focus a phenotypic search there. But, there is good expression of ceh-60 in the pharynx muscle. PM6 surrounds the grinder, and there are some fascinating biology that is going on there (see David Raizen’s 2020 PLoS One paper). Perhaps ceh-60 is involved in grinder formation, or the transdifferaction of surrounding cells, or re-establishing the muscle-nature of PM6 after it has transdifferentiated. If ceh-60 had an important role in grinder formation, I would expect a slow-growth phenotype, but no one has reported that, so maybe there is nothing there.

It is certainly a reasonable hypothesis for expression of ceh-60 in the PM6 muscle cells to be related to pharynx grinder formation; and we agree it would be worth investigating grinder structure in future research - for example using TEM as recently done by David Raizen’s team (2020) - in ceh-60 mutants. We now discuss this in the revised manuscript (lines 525-532):

“CEH-60’s function in the pharynx may not be related to muscle structure directly. Recently, it has been shown that the PM6 pharyngeal muscle cells, in which ceh-60 is abundantly expressed and which surround the pharynx grinder, transdifferentiate into secretory cells during lethargus, possibly aiding in the construction of a new grinder (Sparacio et al. 2020). The grinder is an extracellular matrix, constructed during each larval transition phase, coinciding with earlier-reported cyclic expression of ceh-60 (Hendriks et al. 2014, Van Rompay et al. 2015). CEH-60 may act in PM6 cells to aid in this cyclic transdifferentiation from muscle cells to secretory cells, or in re-establishment of muscle nature once the extracellular matrix of the grinder is built.”

We do not observe an obvious slow growth phenotype in mutants carrying any of the ceh-60 alleles mentioned in our (current and previous) work, although we cannot yet rule out minor differences (few minutes to hours). 

We have addressed concerns about the CEH-60 expression pattern in AWC neurons below, where the question is posed in more detail.

The authors could look at the maceration of fluorescent bacteria for example to quantify grinder efficiency. 

In response to this comment, we performed extra experiments using OP50-dsRed E. coli bacteria and quantified isthmus peristalsis, the “swallowing” motion of the pharynx that moves food from the pharynx corpus to the terminal bulb. We observed no differences between wild-type animals and ceh-60 mutants, strengthening our hypothesis that food intake in these animals is essentially normal. The results are presented in Fig. 4B and lines 501-504 of the revised manuscript:

“Additionally, we quantified isthmus peristalsis, a contraction of pharynx muscles that carries food from the corpus of the pharynx to the terminal bulb, using fluorescent E. coli OP50 bacteria. We observed no difference between wild type and ceh-60 mutant animals (Fig 4.B).”

 The assay is described in the methods section, in lines 203-212 of the revised manuscript.

Should the authors look at other large-scale datasets, they will find that ceh-60 oscillates in a time sequence in larval development that is consistent with a role in grinder formation or some other aspect of the larval development cycle.

Indeed, we have also reported on ceh-60’s transcriptional oscillation pattern in previous work (Van Rompay et al. 2015, Van de Walle et al. 2019), which correlates with molting cycles, hence also with grinder formation. We have updated the manuscript to include this hypothesis, as per details quoted two comments above. 

The body wall muscle targets are very perplexing. If one examines the Cao et al 2017 data in detail, one will see that ceh-60 transcripts are detected at only very low levels (~40X less than in the intestine and ~100X less than in the pharynx muscle). Several of those genes are well-known to be exclusively expressed in non-pharynx muscle (unc-54, myo-3 etc). This, in combination with the lack of vit-gene targets detected, call into question the validity of the experiments (sorry). It's a head scratcher and some things don't add up.

We agree that the data present a “head scratcher”, but not an unsolvable one. All technical parameters indicate that the data quality and analysis are sound, which means that integrating these results with previous knowledge on CEH-60 is in essence a conceptual challenge.

The presence of CEH-60 gene targets in the body wall muscle, where CEH-60 itself is not expressed, does not necessarily pose a problem, because CEH-60 can also act as a repressor of transcription (Dowen 2019). It can therefore be hypothesized that in pharyngeal muscle cells, CEH-60 could act as a repressor of body wall muscle genes such as unc-54 and myo¬-3. To clarify these points, we added the following lines to the revised manuscript (lines 520-524)

“While our dataset of CEH-60 targets also contains several genes known to be expressed in body wall muscle, CEH-60 itself is absent from this tissue (Cao et al. 2017; Van de Walle et al. 2019). Because CEH-60 can also act as a repressor of transcription (Dowen 2019), these data could indicate that in the pharynx, CEH-60 actively represses transcription of certain body wall muscle genes. It would be interesting to explore this further in future research.”

In the revised text, these lines are followed by the pharynx/grinder rationale tying into David Raizen’s work, mentioned three comments back, as a plausible and fascinating route for future investigation (lines 525-532). 

On that note, the Cao et al data shows no expression of ceh-60 in ciliated neurons (which would include AWC) in L2s, which should raise some concern about reporter expression of ceh-60 in the AWC being a result of transgene artifact. (Sorry, but its not clear if the authors are making a claim that ceh-60 is expressed in the AWCs only in adults; if so, then the Cao data would not necessarily be relevant.

Different fluorescent reporter constructs made by us and others (Reece-Hoyes et al. 2007, Van Rompay et al. 2015, Dowen et al 2019, Van de Walle et al. 2019) all point towards expression of ceh-60 in amphid neurons. These include transcriptional, translational and fosmid-based constructs with different fluorescent markers inserted at different points in the sequence, and driven by different promoter variants. In our experience, AWC expression is present throughout life. This means that there is indeed a disagreement between all these reporters and the Cao et al. curated dataset, where ceh-60 was not assigned to the ciliated neuron cluster. However, many technical reasons may equally well explain ceh-60’s transcript absence in the latter, and unfortunately, no other single-cell datasets are available that contain any ceh-60 data at all (Lorenzo et al. 2020, Nucleic Acids Res., Hammerlund et al. 2018, Neuron). The ceh-60 locus is rather complex, containing an internal transcription start site and several overlapping possible transcripts, which have not yet been teased out. Clearly, the last ink has far from been spilled on a possible role for CEH-60 in these cells, but we uphold that evidence at present argues for the presence of this transcription factor in AWC. 

The authors might want to cross-reference their list of targets (as revealed by DamID or ChIP) with theose genes that are co-expressed with ceh-20 in the pharynx muscle. These would be high-value targets and worth further discussion.

We are unsure what the reviewer means by this. As far as we know, there is no list of CEH-20 gene targets in pharynx muscle nor a list of pharyngeal genes differentially expressed on ceh-20 knockout. We agree that, should a list of pharyngeal targets of CEH-20 become available, cross-referencing it with our CEH-60 pharyngeal muscle targets would indeed be very interesting. 

Minor point: On line 280, it is not clear what the authors mean by writing, ‘9 are specifically expressed in pharyngeal muscle’. Certainly, they must not mean ‘exclusively expressed in pharynx muscle’, since those genes are known to be expressed in BWMs. Perhaps they mean that there is good evidence that they are also expressed in the pharynx. If so, they should clarify that sentence.

We apologize for the confusion. The lines now read (lines 282-284 in the revised manuscript): 

“Indeed, for 9 out of the 23 genes classified under “muscle structure development”, the largest muscle-related biological process in the GO analysis (Fig 1), there is evidence for expression in pharyngeal muscle.”

In addition, we made following small adjustments:

- Throughout the manuscript and in Fig S1: ceh-60::dam � dam::ceh-60

 This was done to correctly reflect the order of the sequences in the fusion construct.

- Line 223-224: the dam::ceh-60 read number � the normalized dam::ceh-60 read number.

- Line 224: read number � normalized read number.

---

## [Decision Letter · Decision Letter 1]

12 Nov 2020

DamID identifies targets of CEH-60/PBX that are associated with neuron development and muscle structure in Caenorhabditis elegans

PONE-D-20-27700R1

Dear Dr. Van de Walle,

We’re pleased to inform you that your manuscript has been judged scientifically suitable for publication and will be formally accepted for publication once it meets all outstanding technical requirements.

Kind regards,

Denis Dupuy, Ph.D.

Academic Editor

PLOS ONE

Additional Editor Comments (optional):

Reviewers' comments:

Reviewer's Responses to Questions

**Comments to the Author**

1. If the authors have adequately addressed your comments raised in a previous round of review and you feel that this manuscript is now acceptable for publication, you may indicate that here to bypass the “Comments to the Author” section, enter your conflict of interest statement in the “Confidential to Editor” section, and submit your "Accept" recommendation.

Reviewer #1: All comments have been addressed

Reviewer #2: All comments have been addressed

2. Is the manuscript technically sound, and do the data support the conclusions?

Reviewer #1: Yes

Reviewer #2: (No Response)

3. Has the statistical analysis been performed appropriately and rigorously? 

Reviewer #1: Yes

Reviewer #2: (No Response)

4. Have the authors made all data underlying the findings in their manuscript fully available?

Reviewer #1: Yes

Reviewer #2: (No Response)

5. Is the manuscript presented in an intelligible fashion and written in standard English?

Reviewer #1: Yes

Reviewer #2: (No Response)

6. Review Comments to the Author

Reviewer #1: All concerns raised during peer review have been adequately addressed. I feel that the paper is now ready for publication.

Reviewer #2: (No Response)

7. PLOS authors have the option to publish the peer review history of their article (what does this mean?). If published, this will include your full peer review and any attached files.

Reviewer #1: No

Reviewer #2: No

---

## [Editor Report · Acceptance letter]

23 Nov 2020

PONE-D-20-27700R1 

DamID identifies targets of CEH-60/PBX that are associated with neuron development and muscle structure in *Caenorhabditis elegans*

Dear Dr. Van de Walle:

I'm pleased to inform you that your manuscript has been deemed suitable for publication in PLOS ONE. Congratulations! Your manuscript is now with our production department. 

Kind regards, 

on behalf of

Dr. Denis Dupuy 

Academic Editor

PLOS ONE